# Deep Evidential Regression


Alexander Amini[1], Wilko Schwarting[1], Ava Soleimany[2], Daniela Rus[1]
[1] Computer Science and Artificial Intelligence Lab (CSAIL),
Massachusetts Institute of Technology (MIT)
[2] Harvard Graduate Program in Biophysics



## Abstract

Deterministic neural networks (NNs) are increasingly being deployed in safety critical domains, where calibrated, robust, and efficient measures of uncertainty are crucial. In this paper, we propose a novel method for training non-Bayesian NNs to estimate a continuous target as well as its associated evidence in order to learn both aleatoric and epistemic uncertainty. We accomplish this by placing evidential priors over the original Gaussian likelihood function and training the NN to infer the hyperparameters of the evidential distribution. We additionally impose priors during training such that the model is regularized when its predicted evidence is not aligned with the correct output. Our method does not rely on sampling during inference or on out-of-distribution (OOD) examples for training, thus enabling efficient and scalable uncertainty learning. We demonstrate learning well-calibrated measures of uncertainty on various benchmarks, scaling to complex computer vision tasks, as well as robustness to adversarial and OOD test samples.


## 1 Introduction

Regression-based neural networks (NNs) are being deployed in safety critical domains in computer vision [15] as well as in robotics and control [1, 6], where the ability to infer model uncertainty is crucial for eventual wide-scale adoption. Furthermore, precise and calibrated uncertainty estimates are useful for interpreting confidence, capturing domain shift of out-of-distribution (OOD) test samples, and recognizing when the model is likely to fail.

Figure 1: **Evidential regression** simultaneously learns a continuous target along with aleatoric (data) and epistemic (model) uncertainty. Given an input, the network is trained to predict the parameters of an evidential distribution, which models a higher-order probability distribution over the individual likelihood parameters, $(\mu, \sigma^2)$.

There are two axes of NN uncertainty that can be modeled: (1) uncertainty in the data, called aleatoric uncertainty, and (2) uncertainty in the prediction, called epistemic uncertainty. While representations of aleatoric uncertainty can be learned directly from data, there exist several approaches for estimating epistemic uncertainty, such as Bayesian NNs, which place probabilistic priors over network weights and use sampling to approximate output variance [25].

However, Bayesian NNs face several limitations, including the intractability of directly inferring the posterior distribution of the weights given data, the requirement and computational expense of sampling during inference, and the question of how to choose a weight prior.



In contrast, evidential deep learning formulates learning as an evidence acquisition process [42, 32]. Every training example adds support to a learned higher-order, *evidential* distribution. Sampling from this distribution yields instances of lower-order likelihood functions from which the data was drawn. Instead of placing priors on network weights, as is done in Bayesian NNs, evidential approaches place priors directly over the likelihood function. By training a neural network to output the hyperparameters of the higher-order evidential distribution, a grounded representation of both epistemic and aleatoric uncertainty can then be learned without the need for sampling.

To date, evidential deep learning has been targeted towards discrete classification problems [42, 32, 22] and has required either a well-defined distance measure to a maximally uncertain prior [42] or relied on training with OOD data to inflate model uncertainty [32, 31]. In contrast, continuous regression problems present the complexity of lacking a well-defined distance measure to regularize the inferred evidential distribution. Further, pre-defining a reasonable OOD dataset is non-trivial in the majority of applications; thus, methods to obtain calibrated uncertainty on OOD data from only an in-distribution training set are required.

We present a novel approach that models the uncertainty of regression networks via learned evidential distributions (Fig. 1). Specifically, this work makes the following contributions:

1. A novel and scalable method for learning epistemic and aleatoric uncertainty on regression problems, without sampling during inference or training with out-of-distribution data;
2. Formulation of an evidential regularizer for continuous regression problems, necessary for penalizing incorrect evidence on errors and OOD examples;
3. Evaluation of epistemic uncertainty on benchmark and complex vision regression tasks along with comparisons to state-of-the-art NN uncertainty estimation techniques; and
4. Robustness and calibration evaluation on OOD and adversarially perturbed test input data.

## 2 Modelling uncertainties from data

### 2.1 Preliminaries

Consider the following supervised optimization problem: given a dataset, $\mathcal{D}$, of $N$ paired training examples, $\mathcal{D} = \{x_i, y_i\}_{i=1}^N$, we aim to learn a functional mapping $f$, parameterized by a set of weights, $w$, which approximately solves the following optimization problem:

$$\min_{w} J(w); \quad J(w) = \frac{1}{N} \sum_{i=1}^N \mathcal{L}_i(w), \tag{1}$$

where $\mathcal{L}_i(\cdot)$ describes a loss function. In this work, we consider deterministic regression problems, which commonly optimize the sum of squared errors, $\mathcal{L}_i(w) = \frac{1}{2} \|y_i - f(x_i; w)\|^2$. In doing so, the model is encouraged to learn the average correct answer for a given input, but does not explicitly model any underlying noise or uncertainty in the data when making its estimation.

### 2.2 Maximum likelihood estimation

One can approach this problem from a maximum likelihood perspective, where we learn model parameters that maximize the likelihood of observing a particular set of training data. In the context of deterministic regression, we assume our targets, $y_i$, were drawn i.i.d. from a distribution such as a Gaussian with mean and variance parameters $\boldsymbol{\theta} = (\mu, \sigma^2)$. In maximum likelihood estimation (MLE), we aim to learn a model to infer $\boldsymbol{\theta}$ that maximize the likelihood of observing our targets, $y$, given by $p(y_i|\boldsymbol{\theta})$. This is achieved by minimizing the negative log likelihood loss function:

$$\mathcal{L}_i(w) = -\log p(y_i | \underbrace{\mu, \sigma^2}_{\boldsymbol{\theta}}) = \frac{1}{2} \log(2\pi\sigma^2) + \frac{(y_i - \mu)^2}{2\sigma^2}. \tag{2}$$

In learning $\boldsymbol{\theta}$, this likelihood function successfully models the uncertainty in the data, also known as the aleatoric uncertainty. However, our model is oblivious to its predictive epistemic uncertainty [25].

In this paper, we present a novel approach for estimating the evidence supporting network predictions in regression by directly learning both the aleatoric uncertainty present in the data as well as the

2

Figure 2: **Normal Inverse-Gamma distribution.** Different realizations of our evidential distribution (A) correspond to different levels of confidences in the parameters (e.g. $\mu, \sigma^2$). Sampling from a single realization of a higher-order evidential distribution (B), yields lower-order likelihoods (C) over the data (e.g. $p(y|\mu,\sigma^2)$). Darker shading indicates higher probability mass. We aim to learn a model that predicts the target, $y$, from an input, $x$, with an evidential prior imposed on our likelihood to enable uncertainty estimation.

model's underlying epistemic uncertainty. We achieve this by placing higher-order prior distributions over the learned parameters governing the distribution from which our observations are drawn.

## 3 Evidential uncertainty for regression

### 3.1 Problem setup

We consider the problem where the observed targets, $y_i$, are drawn i.i.d. from a Gaussian distribution, as in standard MLE (Sec. 2.2), but now with *unknown mean and variance* $(\mu, \sigma^2)$, which we seek to also probabilistically estimate. We model this by placing a prior distribution on $(\mu, \sigma^2)$. If we assume observations are drawn from a Gaussian, in line with assumptions Sec. 2.2, this leads to placing a Gaussian prior on the unknown mean and an Inverse-Gamma prior on the unknown variance:

$$(y_1, \ldots, y_N) \sim \mathcal{N}(\mu, \sigma^2)$$
$$\mu \sim \mathcal{N}(\gamma, \sigma^2 v^{-1}) \qquad \sigma^2 \sim \Gamma^{-1}(\alpha, \beta). \tag{3}$$

where $\Gamma(\cdot)$ is the gamma function, $\boldsymbol{m} = (\gamma, v, \alpha, \beta)$, and $\gamma \in \mathbb{R}$, $v > 0$, $\alpha > 1$, $\beta > 0$.

Our aim is to estimate a posterior distribution $q(\mu, \sigma^2) = p(\mu, \sigma^2|y_1, \ldots, y_N)$. To obtain an approximation for the true posterior, we assume that the estimated distribution can be factorized [39] such that $q(\mu, \sigma^2) = q(\mu)\,q(\sigma^2)$. Thus, our approximation takes the form of the Gaussian conjugate prior, the Normal Inverse-Gamma (`NIG`) distribution:

$$p(\underbrace{\mu, \sigma^2}_{\boldsymbol{\theta}} \,|\, \underbrace{\gamma, v, \alpha, \beta}_{\boldsymbol{m}}) = \frac{\beta^\alpha \sqrt{v}}{\Gamma(\alpha)\sqrt{2\pi\sigma^2}} \left(\frac{1}{\sigma^2}\right)^{\alpha+1} \exp\left\{-\frac{2\beta + v(\gamma - \mu)^2}{2\sigma^2}\right\}. \tag{4}$$

A popular interpretation of the parameters of this conjugate prior distribution is in terms of "virtual-observations" in support of a given property [23]. For example, the mean of a `NIG` distribution can be intuitively interpreted as being estimated from $v$ virtual-observations with sample mean $\gamma$, while its variance is estimated from $\alpha$ virtual-observations with sample mean $\gamma$ and sum of squared deviations $2v$. Following from this interpretation, we define the total evidence, $\Phi$, of our evidential distributions as the sum of all inferred virtual-observations counts: $\Phi = 2v + \alpha$.

Drawing a sample $\boldsymbol{\theta}_j$ from the `NIG` distribution yields a single instance of our likelihood function, namely $\mathcal{N}(\mu_j, \sigma_j^2)$. Thus, the `NIG` hyperparameters, $(\gamma, v, \alpha, \beta)$, determine not only the location but also the dispersion concentrations, or uncertainty, associated with our inferred likelihood function. Therefore, we can interpret the `NIG` distribution as the *higher-order*, *evidential* distribution on top of the unknown *lower-order* likelihood distribution from which observations are drawn.

3

For example, in Fig. 2A we visualize different evidential `NIG` distributions with varying model parameters. We illustrate that by increasing the evidential parameters (i.e. $v, \alpha$) of this distribution, the p.d.f. becomes tightly concentrated about its inferred likelihood function. Considering a single parameter realization of this higher-order distribution (Fig. 2B), we can subsequently sample many lower-order realizations of our likelihood function, as shown in Fig. 2C.

In this work, we use neural networks to infer, given an input, the hyperparameters, $m$, of this higher-order, evidential distribution. This approach presents several distinct advantages compared to prior work. First, our method enables simultaneous learning of the desired regression task, along with aleatoric and epistemic uncertainty estimation, by enforcing evidential priors and without leveraging any out-of-distribution data during training. Second, since the evidential prior is a higher-order `NIG` distribution, the maximum likelihood Gaussian can be computed analytically from the expected values of the $(\mu, \sigma^2)$ parameters, without the need for sampling. Third, we can effectively estimate the epistemic or model uncertainty associated with the network's prediction by simply evaluating the variance of our inferred evidential distribution.

### 3.2 Prediction and uncertainty estimation

The aleatoric uncertainty, also referred to as statistical or data uncertainty, is representative of unknowns that differ each time we run the same experiment. The epistemic (or model) uncertainty, describes the estimated uncertainty in the prediction. Given a `NIG` distribution, we can compute the prediction, aleatoric, and epistemic uncertainty as

$$\underbrace{\mathbb{E}[\mu] = \gamma}_{\text{prediction}}, \qquad \underbrace{\mathbb{E}[\sigma^2] = \tfrac{\beta}{\alpha-1}}_{\text{aleatoric}}, \qquad \underbrace{\text{Var}[\mu] = \tfrac{\beta}{v(\alpha-1)}}_{\text{epistemic}}. \tag{5}$$

Complete derivations for these moments are available in Sec. S1.1. Note that $\text{Var}[\mu] = \mathbb{E}[\sigma^2]/v$, which is expected as $v$ is one of our two evidential virtual-observation counts.

### 3.3 Learning the evidential distribution

Having formalized the use of an evidential distribution to capture both aleatoric and epistemic uncertainty, we next describe our approach for learning a model to output the hyperparameters of this distribution. For clarity, we structure the learning process as a multi-task learning problem, with two distinct parts: (1) acquiring or maximizing model evidence in support of our observations and (2) minimizing evidence or inflating uncertainty when the prediction is wrong. At a high level, we can think of (1) as a way of fitting our data to the evidential model while (2) enforces a prior to remove incorrect evidence and inflate uncertainty.

**(1) Maximizing the model fit.** From Bayesian probability theory, the "model evidence", or marginal likelihood, is defined as the likelihood of an observation, $y_i$, given the evidential distribution parameters $m$ and is computed by marginalizing over the likelihood parameters $\theta$:

$$p(y_i|\boldsymbol{m}) = \frac{p(y_i|\boldsymbol{\theta}, \boldsymbol{m})p(\boldsymbol{\theta}|\boldsymbol{m})}{p(\boldsymbol{\theta}|y_i, \boldsymbol{m})} = \int_{\sigma^2=0}^{\infty} \int_{\mu=-\infty}^{\infty} p(y_i|\mu, \sigma^2)p(\mu, \sigma^2|\boldsymbol{m}) \, \mathrm{d}\mu \, \mathrm{d}\sigma^2 \tag{6}$$

The model evidence is, in general, not straightforward to evaluate since computing it involves integrating out the dependence on latent model parameters. However, in the case of placing a `NIG` evidential prior on our Gaussian likelihood function an analytical solution does exist:

$$p(y_i|\boldsymbol{m}) = \text{St}\left(y_i; \gamma, \frac{\beta(1+v)}{v\,\alpha}, 2\alpha\right). \tag{7}$$

where $\text{St}\left(y; \mu_{\text{St}}, \sigma^2_{\text{St}}, v_{St}\right)$ is the Student-t distribution evaluated at $y$ with location $\mu_{\text{St}}$, scale $\sigma^2_{\text{St}}$, and $v_{St}$ degrees of freedom. We denote the loss, $\mathcal{L}_i^{\text{NLL}}(\boldsymbol{w})$, as the negative logarithm of model evidence

$$\mathcal{L}_i^{\text{NLL}}(\boldsymbol{w}) = \tfrac{1}{2}\log\left(\tfrac{\pi}{v}\right) - \alpha\log(\Omega) + \left(\alpha + \tfrac{1}{2}\right)\log((y_i - \gamma)^2 v + \Omega) + \log\left(\tfrac{\Gamma(\alpha)}{\Gamma(\alpha+\tfrac{1}{2})}\right) \tag{8}$$

where $\Omega = 2\beta(1+v)$. Complete derivations for Eq. 7 and Eq. 8 are provided in Sec. S1.2. This loss provides an objective for training a NN to output parameters of a `NIG` distribution to fit the observations by maximizing the model evidence.



**(2) Minimizing evidence on errors.** Next, we describe how to regularize training by applying an incorrect evidence penalty (i.e., high uncertainty prior) to try to minimize evidence on incorrect predictions. This has been demonstrated with success in the classification setting where non-misleading evidence is removed from the posterior, and the uncertain prior is set to a uniform Dirichlet [42]. The analogous minimization in the regression setting involves $KL[\,p(\boldsymbol{\theta}|\boldsymbol{m})\,||\,p(\boldsymbol{\theta}|\tilde{\boldsymbol{m}})\,]$, where $\tilde{\boldsymbol{m}}$ are the parameters of the uncertain `NIG` prior with zero evidence (i.e., $\{\alpha, \upsilon\} = 0$). Unfortunately, the KL between any `NIG` and the zero evidence `NIG` prior is undefined[(1)]. Furthermore, this loss should not be enforced everywhere, but instead specifically where the posterior is "misleading". Past works in classification [42] accomplish this by using the ground truth likelihoood classification (the one-hot encoded labels) to remove "non-misleading" evidence. However, in regression, it is not possible to penalize evidence everywhere except our single label point estimate, as this space is infinite and unbounded. Thus, these previous approaches for regularizing evidential learning are not applicable.

To address these challenges in the regression setting, we formulate a novel evidence regularizer, $\mathcal{L}_i^{\text{R}}$, scaled on the error of the $i$-th prediction,

$$\mathcal{L}_i^{\text{R}}(\boldsymbol{w}) = |y_i - \mathbb{E}[\mu_i]| \cdot \Phi = |y_i - \gamma| \cdot (2\upsilon + \alpha). \tag{9}$$

This loss imposes a penalty whenever there is an error in the prediction and scales with the total evidence of our inferred posterior. Conversely, large amounts of predicted evidence will not be penalized as long as the prediction is close to the target. A naïve alternative to directly penalizing evidence would be to soften the zero-evidence prior to instead have $\epsilon$-evidence such that the KL is finite and defined. However, doing so results in hypersensitivity to the selection of $\epsilon$, as it should be small yet $KL \to \infty$ as $\epsilon \to 0$. We demonstrate the added value of our evidential regularizer through ablation analysis (Sec. 4.1), the limitations of the soft KL regularizer (Sec. S2.1.3), and the ability to learn disentangled aletoric and epistemic uncertainty (Sec. S2.1.4).

**Summary and implementation details.** The total loss, $\mathcal{L}_i(\boldsymbol{w})$, consists of the two loss terms for maximizing and regularizing evidence, scaled by a regularization coefficient, $\lambda$,

$$\mathcal{L}_i(\boldsymbol{w}) = \mathcal{L}_i^{\text{NLL}}(\boldsymbol{w}) + \lambda\, \mathcal{L}_i^{\text{R}}(\boldsymbol{w}). \tag{10}$$

Here, $\lambda$ trades off uncertainty inflation with model fit. Setting $\lambda = 0$ yields an over-confident estimate while setting $\lambda$ too high results in over-inflation[(2)]. In practice, our NN is trained to output the parameters, $\boldsymbol{m}$, of the evidential distribution: $\boldsymbol{m}_i = f(\boldsymbol{x}_i; \boldsymbol{w})$. Since $\boldsymbol{m}$ is composed of 4 parameters, $f$ has 4 output neurons for every target $y$. We enforce the constraints on $(\upsilon, \alpha, \beta)$ with a `softplus` activation (and additional +1 added to $\alpha$ since $\alpha > 1$). Linear activation is used for $\gamma \in \mathbb{R}$.

## 4 Experiments

### 4.1 Predictive accuracy and uncertainty benchmarking

We first qualitatively compare the performance of our approach against a set of baselines on a one-dimensional cubic regression dataset (Fig. 3). Following [20, 28], we train models on $y = x^3 + \epsilon$, where $\epsilon \sim \mathcal{N}(0, 3)$ within $\pm 4$ and test within $\pm 6$. We compare aleatoric (A) and epistemic (B) uncertainty estimation for baseline methods (left), evidence without regularization (middle), and with regularization (right). Gaussian MLE [36] and Ensembling [28] are used as respective baseline methods. All aleatoric methods (A) accurately capture uncertainty within the training distribution, as expected. Epistemic uncertainty (B) captures uncertainty on OOD data; our proposed evidential method estimates uncertainty appropriately and grows on OOD data, without dependence on sampling. Training details and additional experiments for this example are available in Sec. S2.1.

Additionally, we compare our approach to baseline methods for NN predictive uncertainty estimation on real world datasets used in [20, 28, 9]. We evaluate our proposed evidential regression method against results presented for model ensembles [28] and dropout [9] based on root mean squared error (RMSE), negative log-likelihood (NLL), and inference speed. Table 1 indicates that even though, unlike the competing approaches, the loss function for evidential regression does not explicitly optimize accuracy, it remains competitive with respect to RMSE while being the top performer on all datasets for NLL and speed. To give the two baseline methods maximum advantage, we parallelize

---

[(1)]Please refer to Sec. S1.3 for derivation of the KL between two `NIG`s, along with a no-evidence `NIG` prior.
[(2)]Experiments demonstrating the effect of $\lambda$ on a learning problem are provided in Sec. S2.1.3

5

| | RMSE | | | NLL | | | Inference Speed (ms) | | |
|---|---|---|---|---|---|---|---|---|---|
| Dataset | Dropout | Ensembles | Evidential | Dropout | Ensembles | Evidential | Dropout | Ensemble | Evidential |
| Boston | **2.97 ± 0.19** | 3.28 ± 1.00 | **3.06 ± 0.16** | 2.46 ± 0.06 | 2.41 ± 0.25 | **2.35 ± 0.06** | 3.24 | 3.35 | **0.85** |
| Concrete | **5.23 ± 0.12** | 6.03 ± 0.58 | 5.85 ± 0.15 | 3.04 ± 0.02 | 3.06 ± 0.18 | **3.01 ± 0.02** | 2.99 | 3.43 | **0.94** |
| Energy | **1.66 ± 0.04** | 2.09 ± 0.29 | 2.06 ± 0.10 | 1.99 ± 0.02 | **1.38 ± 0.22** | **1.39 ± 0.06** | 3.08 | 3.80 | **0.87** |
| Kin8nm | 0.10 ± 0.00 | **0.09 ± 0.00** | **0.09 ± 0.00** | -0.95 ± 0.01 | -1.20 ± 0.02 | **-1.24 ± 0.01** | 3.24 | 3.79 | **0.97** |
| Naval | 0.01 ± 0.00 | **0.00 ± 0.00** | **0.00 ± 0.00** | -3.80 ± 0.01 | **-5.63 ± 0.05** | **-5.73 ± 0.07** | 3.31 | 3.37 | **0.84** |
| Power | **4.02 ± 0.04** | 4.11 ± 0.17 | 4.23 ± 0.09 | **2.80 ± 0.01** | **2.79 ± 0.04** | **2.81 ± 0.07** | 2.93 | 3.36 | **0.85** |
| Protein | **4.36 ± 0.01** | 4.71 ± 0.06 | 4.64 ± 0.03 | 2.89 ± 0.00 | 2.83 ± 0.02 | **2.63 ± 0.00** | 3.45 | 3.68 | **1.18** |
| Wine | 0.62 ± 0.01 | 0.64 ± 0.04 | **0.61 ± 0.02** | 0.93 ± 0.01 | 0.94 ± 0.12 | **0.89 ± 0.05** | 3.00 | 3.32 | **0.86** |
| Yacht | **1.11 ± 0.09** | 1.58 ± 0.48 | 1.57 ± 0.56 | 1.55 ± 0.03 | 1.18 ± 0.21 | **1.03 ± 0.19** | 2.99 | 3.36 | **0.87** |

Table 1: **Benchmark regression tests.** RMSE, negative log-likelihood (NLL), and inference speed for dropout sampling [9], model ensembling [28], and evidential regression. Top scores for each metric and dataset are bolded (within statistical significance), $n = 5$ for sampling baselines. Evidential models outperform baseline methods for NLL and inference speed on all datasets.

their sampled inference ($n = 5$). Dropout requires additional multiplications with the sampled mask, resulting in slightly slower inference compared to ensembles, whereas evidence only requires a single forward pass and network. Training details for Table 1 are available in Sec. S2.2.

### 4.2 Monocular depth estimation

After establishing benchmark comparison results, in this subsection we demonstrate the scalability of our evidential learning approach by extending it to the complex, high-dimensional task of depth estimation. Monocular end-to-end depth estimation is a central problem in computer vision and involves learning a representation of depth directly from an RGB image of the scene. This is a challenging learning task as the target $y$ is very high-dimensional, with predictions at every pixel.

Our training data consists of over 27k RGB-to-depth, $H \times W$, image pairs of indoor scenes (e.g. kitchen, bedroom, etc.) from the NYU Depth v2 dataset [35]. We train a U-Net style NN [41] for inference and test on a disjoint test-set of scenes[(3)]. The final layer outputs a single $H \times W$ activation map in the case of vanilla regression, dropout, and ensembling. Spatial dropout uncertainty sampling [2, 45] is used for the dropout implementation. Evidential regression outputs four of these output maps, corresponding to $(\gamma, \upsilon, \alpha, \beta)$, with constraints according to Sec. 3.3.

Figure 3: **Toy uncertainty estimation.** Aleatoric (A) and epistemic (B) uncertainty estimates on the dataset $y = x^3 + \epsilon$, $\epsilon \sim \mathcal{N}(0, 3)$. Regularized evidential regression (right) enables precise prediction within the training regime and conservative epistemic uncertainty estimates in regions with no training data. Baseline results are also illustrated.

We evaluate the models in terms of their accuracy and their predictive epistemic uncertainty on unseen test data. Fig. 4A visualizes the predicted depth, absolute error from ground truth, and predictive entropy across two randomly picked test images. Ideally, a strong epistemic uncertainty measure would capture errors in the prediction (i.e., roughly correspond to where the model is making errors). Compared to dropout and ensembling, evidential modeling captures the depth errors while providing

Figure 4: **Epistemic uncertainty in depth estimation.** (A) Example pixel-wise depth predictions and uncertainty for each model. (B) Relationship between prediction confidence level and observed error; a strong inverse trend is desired. (C) Model uncertainty calibration [27]; (ideal: $y = x$). Inset shows calibration errors.

---

[(3)]Full dataset, model, training, and performance details for depth models are available in Sec. S3.

6

Figure 5: **Uncertainty on out-of-distribution (OOD) data.** Evidential models estimate low uncertainty (entropy) on in-distribution (ID) data and inflate uncertainty on OOD data. (A) Cumulative density function (CDF) of ID and OOD entropy for tested methods. OOD detection assessed via AUC-ROC. (B) Uncertainty (entropy) comparisons across methods. (C) Full density histograms of entropy estimated by evidential regression on ID and OOD data, along with sample images (D). All data has not been seen during training.

clear and localized predictions of confidence. In general, dropout drastically underestimates the amount of uncertainty present, while ensembling occasionally overestimates the uncertainty. Fig. 4B shows how each model performs as pixels with uncertainty greater than certain thresholds are removed. Evidential models exhibit strong performance, as error steadily decreases with increasing confidence.

Fig. 4C additionally evaluates the calibration of our uncertainty estimates. Calibration curves are computed according to [27], and ideally follows $y = x$ to represent, for example, that a target falls in a 90% confidence interval approximately 90% of the time. Again, we see that dropout overestimates confidence when considering low confidence scenarios (calibration error: $0.126$). Ensembling exhibits better calibration error ($0.048$) but is still outperformed by the proposed evidential method ($0.033$). Results show evaluations from multiple trials, with individual trials available in Sec. S3.3.

In addition to epistemic uncertainty experiments, we also evaluate aleatoric uncertainty estimates, with comparisons to Gaussian MLE learning. Since evidential models fit the data to a higher-order Gaussian distribution, it is expected that they can accurately learn aleatoric uncertainty (as is also shown in [42, 18]). Therefore, we present these aleatoric results in Sec. S3.4 and focus the remainder of the results on evaluating the harder task of epistemic uncertainty estimation in the context of out-of-distribution (OOD) and adversarily perturbed samples.

### 4.3 Out-of distribution testing

A key use of uncertainty estimation is to understand when a model is faced with test samples that fall out-of-distribution (OOD) or when the model's output cannot be trusted. In this subsection, we investigate the ability of evidential models to capture increased epistemic uncertainty on OOD data, by testing on images from ApolloScape [21], an OOD dataset of diverse outdoor driving. It is crucial to note here that related methods such as Prior Networks in classification [32, 33] explicitly require OOD data during training to supervise instances of high uncertainty. Our evidential method, like Bayesian NNs, does not have this limitation and sees only in distribution (ID) data during training.

For each method, we feed in the ID and OOD test sets and record the mean predicted entropy for every test image. Fig. 5A shows the cumulative density function (CDF) of entropy for each of the methods and test sets. A distinct positive shift in the entropy CDFs can be seen for evidential models on OOD data and is competitive across methods. Fig. 5B summarizes these entropy distributions as interquartile boxplots to again show clear separation in the uncertainty distribution on OOD data. We focus on the distribution from our evidential models in Fig. 5C and provide sample predictions (ID and OOD) in Fig. 5D. These results show that evidential models, without training on OOD data, capture increased uncertainty on OOD data on par with epistemic uncertainty estimation baselines.

#### 4.3.1 Robustness to adversarial samples

Next, we consider the extreme case of OOD detection where the inputs are adversarially perturbed to inflict error on the predictions. We compute adversarial perturbations to our test set using the Fast Gradient Sign Method (FGSM) [16], with increasing scales, $\epsilon$, of noise. Note that the purpose of this experiment is not to propose a defense for state-of-the-art adversarial attacks, but rather to demonstrate that evidential models accurately capture increased predictive uncertainty on samples which have been adversarially perturbed. Fig. 6A confirms that the absolute error of all methods increases as adversarial noise is added. We also observe a positive effect of noise on our predictive

7

Figure 6: **Evidential robustness under adversarial noise.** Relationship between adversarial noise $\epsilon$ and predictive error (A) and estimated epistemic uncertainty (B). (C) CDF of entropy estimated by evidential regression under the presence of increasing $\epsilon$. (D) Visualization of the effects of increasing adversarial pertubation on the predictions, error, and uncertainty for evidential regression. Results of sample test-set image are shown.

uncertainty estimates in Fig. 6B. Furthermore, we observe that the entropy CDF steadily shifts towards higher uncertainties as the noise in the input sample increases (Fig. 6C).

The robustness of evidential uncertainty against adversarial perturbations is visualized in greater detail in Fig. 6D, which illustrates the predicted depth, error, and estimated pixel-wise uncertainty as we perturb the input image with greater amounts of noise (left to right). Not only does the predictive uncertainty steadily increase with increasing noise, but the spatial concentrations of uncertainty throughout the image also maintain tight correspondence with the error.

## 5 Related work

Our work builds on a large history of uncertainty estimation [25, 38, 37, 19] and modelling probability distributions using NNs [36, 4, 14, 26].

**Prior networks and evidential models.** A large focus within Bayesian inference is on placing prior distributions over hierarchical models to estimate uncertainty [12, 13]. Our methodology closely relates to evidential deep learning [42] and Prior Networks [32, 33] which place Dirichlet priors over discrete classification predictions. However, these works either rely on regularizing divergence to a fixed, well-defined prior [42, 46], require OOD training data [32, 31, 7, 19], or can only estimate aleatoric uncertainty by performing density estimation [11, 18]. Our work tackles these limitations with focus on continuous regression learning tasks where this divergence regularizer is not well-defined, without requiring any OOD training data to estimate both aleatoric and epistemic uncertainty.

**Bayesian deep learning.** In Bayesian deep learning, priors are placed over network weights that are estimated using variational inference [26]. Approximations via dropout [9, 34, 10, 2], ensembling [28, 40] or other approaches [5, 20] rely on expensive samples to estimate predictive variance. In contrast, we train a deterministic NN to place uncertainty priors over the predictive distribution, requiring only a single forward pass to estimate uncertainty. Additionally, our approach of uncertainty estimation proved to be well calibrated and was capable of detecting OOD and adversarial data.

## 6 Conclusions, limitations, and scope

In this paper, we develop a novel method for learning uncertainty in regression problems by placing evidential priors over the likelihood output. We demonstrate combined prediction with aleatoric and epistemic uncertainty estimation, scalability to complex vision tasks, and calibrated uncer-

8

tainty on OOD data. This method is widely applicable across regression tasks including temporal forecasting [17], property prediction [8], and control learning [1, 30]. While our method presents several advantages over existing approaches, its primary limitations are in tuning the regularization coefficient and in effectively removing non-misleading evidence when calibrating the uncertainty. While dual-optimization formulations [47] could be explored for balancing regularization, we believe further investigation is warranted to discover alternative ways to remove non-misleading evidence. Future analysis using other choices of the variance prior distribution, such as the log-normal or the heavy-tailed log-Cauchy distribution, will be critical to determine the effects of the choice of prior on the estimated likelihood parameters. The efficiency, scalablity, and calibration of our approach could enable the precise and fast uncertainty estimation required for robust NN deployment in safety-critical prediction domains.

## Broader Impact

Uncertainty estimation for neural networks has very significant societal impact. Neural networks are increasingly being trained as black-box predictors and being placed in larger decision systems where errors in their predictions can pose immediate threat to downstream tasks. Systematic methods for calibrated uncertainty estimation under these conditions are needed, especially as these systems are deployed in safety critical domains, such for autonomous vehicle control [29], medical diagnosis [43], or in settings with large dataset imbalances and bias such as crime forecasting [24] and facial recognition [3].

This work is complementary to a large portion of machine learning research which is continually pushing the boundaries on neural network precision and accuracy. Instead of solely optimizing larger models for increased performance, our method focuses on how these models can be equipped with the ability to estimate their own confidence. Our results demonstrating superior calibration of our method over baselines are also critical in ensuring that we can place a certain level of trust in these algorithms and in understanding when they say "I don't know".

While there are clear and broad benefits of uncertainty estimation in machine learning, we believe it is also important to recognize potential societal challenges that may arise. With increased performance and uncertainty estimation capabilities, humans will inevitably become increasingly trusting in a model's predictions, as well as its ability to catch dangerous or uncertain decisions before they are executed. Thus, it is important to continue to pursue redundancy in such learning systems to increase the likelihood that mistakes can be caught and corrected independently.

## Acknowledgments and Disclosure of Funding

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

# Supplementary Materials

## S1 Derivations

### S1.1 Normal Inverse-Gamma moments

We assume our data was drawn from a Gaussian with unknown mean and variance, $(\mu, \sigma^2)$. We probabilistically model these parameters, $\theta$, according to:

$$\mu \sim \mathcal{N}(\gamma, \sigma^2 v^{-1}) \tag{S1}$$

$$\sigma^2 \sim \Gamma^{-1}(\alpha, \beta). \tag{S2}$$

Therefore, the prior joint distribution can be written as:

$$p(\underbrace{\mu, \sigma^2}_{\theta} \mid \underbrace{\gamma, v, \alpha, \beta}_{m}) = p(\mu)\, p(\sigma^2) \tag{S3}$$

$$= \mathcal{N}(\gamma, \sigma^2 v^{-1})\, \Gamma^{-1}(\alpha, \beta) \tag{S4}$$

$$= \frac{\beta^\alpha \sqrt{v}}{\Gamma(\alpha)\sqrt{2\pi\sigma^2}} \left(\frac{1}{\sigma^2}\right)^{\alpha+1} \exp\left\{-\frac{2\beta + v(\gamma - \mu)^2}{2\sigma^2}\right\}. \tag{S5}$$

The first order moments of this distribution represent the maximum likelihood prediction as well as uncertainty (both aleatoric and epistemic).

$$\mathbb{E}[\mu] = \int_{\mu=-\infty}^{\infty} \mu\, p(\mu)\, \mathrm{d}\mu = \gamma \tag{S6}$$

$$\mathbb{E}[\sigma^2] = \int_{\sigma^2=0}^{\infty} \sigma^2\, p(\sigma^2)\, \mathrm{d}\sigma^2 \tag{S7}$$

$$= \int_{\sigma=0}^{\infty} \sigma^2\, p(\sigma^2)\, (2\sigma)\, \mathrm{d}\sigma \tag{S8}$$

$$= \frac{\beta}{\alpha - 1}, \qquad \forall\, \alpha > 1 \tag{S9}$$

$$\mathrm{Var}[\mu] = \int_{\mu=-\infty}^{\infty} \mu^2\, p(\mu)\, \mathrm{d}\mu - (\mathbb{E}[\mu])^2 \tag{S10}$$

$$= \gamma^2 - \frac{\sigma^2}{v} - (\mathbb{E}[\mu])^2 \tag{S11}$$

$$= \gamma^2 - \frac{\frac{\beta}{\alpha-1}}{v} - \gamma^2 \tag{S12}$$

$$= \frac{\beta}{v(\alpha - 1)}, \qquad \forall\, \alpha > 1 \tag{S13}$$

In summary,

$$\underbrace{\mathbb{E}[\mu] = \gamma}_{\text{prediction}}, \qquad \underbrace{\mathbb{E}[\sigma^2] = \frac{\beta}{\alpha-1}}_{\text{aleatoric}}, \qquad \underbrace{\mathrm{Var}[\mu] = \frac{\beta}{v(\alpha-1)}}_{\text{epistemic}}. \tag{S14}$$



### S1.2 Model evidence & Type II Maximum Likelihood Loss

In this subsection, we derive the posterior predictive or model evidence (ie. Eq. 7) of a `NIG` distribution. Marginalizing out $\mu$ and $\sigma$ gives our desired result:

$$p(y_i|\boldsymbol{m}) = \int_{\boldsymbol{\theta}} p(y_i|\boldsymbol{\theta})p(\boldsymbol{\theta}|\boldsymbol{m}) \, d\boldsymbol{\theta} \tag{S15}$$

$$= \int_{\sigma^2=0}^{\infty} \int_{\mu=-\infty}^{\infty} p(y_i|\mu,\sigma^2)p(\mu,\sigma^2|\boldsymbol{m}) \, d\mu \, d\sigma^2 \tag{S16}$$

$$= \int_{\sigma^2=0}^{\infty} \int_{\mu=-\infty}^{\infty} p(y_i|\mu,\sigma^2)p(\mu,\sigma^2|\gamma,\upsilon,\alpha,\beta) \, d\mu \, d\sigma^2 \tag{S17}$$

$$= \int_{\sigma^2=0}^{\infty} \int_{\mu=-\infty}^{\infty} \left[\sqrt{\frac{1}{2\pi\sigma^2}} \exp\left\{-\frac{(y_i-\mu)^2}{2\sigma^2}\right\}\right] \tag{S18}$$

$$\left[\frac{\beta^\alpha \sqrt{\upsilon}}{\Gamma(\alpha)\sqrt{2\pi\sigma^2}} \left(\frac{1}{\sigma^2}\right)^{\alpha+1} \exp\left\{-\frac{2\beta+\upsilon(\gamma-\mu)^2}{2\sigma^2}\right\}\right] d\mu \, d\sigma^2 \tag{S19}$$

$$= \int_{\sigma^2=0}^{\infty} \frac{\beta^\alpha \sigma^{-3-2\alpha}}{\sqrt{2\pi}\sqrt{1+1/\upsilon}\Gamma(\alpha)} \exp\left\{-\frac{2\beta+\frac{\upsilon(y_i-\gamma)^2}{1+\upsilon}}{2\sigma^2}\right\} d\sigma^2 \tag{S20}$$

$$= \int_{\sigma=0}^{\infty} \frac{\beta^\alpha \sigma^{-3-2\alpha}}{\sqrt{2\pi}\sqrt{1+1/\upsilon}\Gamma(\alpha)} \exp\left\{-\frac{2\beta+\frac{\upsilon(y_i-\gamma)^2}{1+\upsilon}}{2\sigma^2}\right\} 2\sigma \, d\sigma \tag{S21}$$

$$= \frac{\Gamma(1/2+\alpha)}{\Gamma(\alpha)}\sqrt{\frac{\upsilon}{\pi}} (2\beta(1+\upsilon))^\alpha \left(\upsilon(y_i-\gamma)^2+2\beta(1+\upsilon)\right)^{-(\frac{1}{2}+\alpha)} \tag{S22}$$

$$p(y_i|\boldsymbol{m}) = \text{St}\left(y_i;\gamma,\frac{\beta(1+\upsilon)}{\upsilon\,\alpha},2\alpha\right). \tag{S23}$$

$\text{St}\left(y;\mu_{\text{St}},\sigma_{\text{St}}^2,\upsilon_{St}\right)$ is the Student-t distribution evaluated at $y$ with location parameter $\mu_{\text{St}}$, scale parameter $\sigma_{\text{St}}^2$, and $\upsilon_{\text{St}}$ degrees of freedom. Using this result we can compute the negative log likelihood loss, $\mathcal{L}_i^{\text{NLL}}$, for sample $i$ as:

$$\mathcal{L}_i^{\text{NLL}} = -\log p(y_i|\boldsymbol{m}) \tag{S24}$$

$$= -\log\left(\text{St}\left(y_i;\gamma,\frac{\beta(1+\upsilon)}{\upsilon\,\alpha},2\alpha\right)\right) \tag{S25}$$

$$\mathcal{L}_i^{\text{NLL}} = \tfrac{1}{2}\log\left(\tfrac{\pi}{\upsilon}\right) - \alpha\log(\Omega) + \left(\alpha+\tfrac{1}{2}\right)\log((y-\gamma)^2\upsilon+\Omega) + \log\left(\frac{\Gamma(\alpha)}{\Gamma(\alpha+\frac{1}{2})}\right) \tag{S26}$$

where $\Omega = 2\beta(1+\upsilon)$.

### S1.3 KL-divergence of the Normal Inverse-Gamma

The KL-divergence between two Normal Inverse-Gamma functions is given by [44]:

$$\mathbb{KL}\big(p(\mu,\sigma^2|\gamma_1,\upsilon_1,\alpha_1,\beta_1) \,\|\, p(\mu,\sigma^2|\gamma_2,\upsilon_2,\alpha_2,\beta_2)\big) \tag{S27}$$

$$= \mathbb{KL}\big(\text{NIG}(\gamma_1,\upsilon_1,\alpha_1,\beta_1) \,\|\, \text{NIG}(\gamma_2,\upsilon_2,\alpha_2,\beta_2)\big) \tag{S28}$$

$$= \frac{1}{2}\frac{\alpha_1}{\beta_1}(\mu_1-\mu_2)^2\upsilon_2 + \frac{1}{2}\frac{\upsilon_2}{\upsilon_1} - \frac{1}{2} + \alpha_2\log\left(\frac{\beta_1}{\beta_2}\right) - \log\left(\frac{\Gamma(\alpha_1)}{\Gamma(\alpha_2)}\right) \tag{S29}$$

$$+ (\alpha_1-\alpha_2)\Psi(\alpha_1) - (\beta_1-\beta_2)\frac{\alpha_1}{\beta_1} \tag{S30}$$

$\Gamma(\cdot)$ is the Gamma function and $\Psi(\cdot)$ is the Digamma function. For zero evidence, both $\alpha = 0$ and $\upsilon = 0$. To compute the KL divergence between one `NIG` distribution and another with zero evidence we can set either $\upsilon_2 = \alpha_2 = 0$ (i.e., reverse-KL) in which case, $\Gamma(0)$ is not well defined, or



$v_1 = \alpha_1 = 0$ (i.e. forward-KL) which causes a divide-by-zero error of $v_1$. In either approach, the KL-divergence between an arbitrary NIG and one with zero evidence cannot be evaluated.

Instead, we briefly consider a naive alternative which can be obtained by considering an $\epsilon$ amount of evidence, where $\epsilon$ is a small constant (instead of strictly 0-evidence). This approach yields a well-defined KL-divergence (with fixed $\gamma, \beta$ at the consequence of a hyper-sensitive $\epsilon$ parameter.

$$\mathbb{KL}\big(\texttt{NIG}(\gamma, v, \alpha, \beta) \,||\, \texttt{NIG}(\gamma, \epsilon, 1+\epsilon, \beta)\big) \tag{S31}$$

$$= \frac{1}{2}\frac{1+\epsilon}{v} - \frac{1}{2} - \log\left(\frac{\Gamma(\alpha)}{\Gamma(1+\epsilon)}\right) + (\alpha - (1+\epsilon))\Psi(\alpha) \tag{S32}$$

In Fig. S1.3 we compare the performance of the KL-divergence regularizer compared to our more direct evidence regularizer, for several realizations of the regularization coefficient, $\lambda$. We observed extreme sensitivity to the setting of $\epsilon$ for different datasets such that we could not achieve the desired regularizing effect for any regularization amount, $\lambda$. Unless otherwise stated, all results were obtained using our direct evidence regularizer instead (Eq. 9).

## S2 Benchmark regression tasks

### S2.1 Cubic toy examples

#### S2.1.1 Dataset and experimental setup

The training set consists of training examples drawn from $y = x^3 + \epsilon$, where $\epsilon \sim \mathcal{N}(0, 3)$ in the region $-4 \leq x \leq 4$, whereas the test data is unbounded (we show in the region $-6 \leq x \leq 6$). This problem setup is identical to that presented in [20, 28]. All models consisted of 100 neurons with 3 hidden layers and were trained to convergence. The data presented in Fig. S1 illustrates the estimated epistemic uncertainty and predicted mean accross the entire test set. Sampling based models [5, 9, 28] used $n = 5$ samples. The evidential model used $\lambda = 0.01$. All models were trained with the Adam optimizer $\eta = $5e-3 for 5000 iterations and a batch size of 128.

#### S2.1.2 Baselines

Figure S1: **Epistemic uncertainty estimation baselines** on the dataset $y = x^3 + \epsilon, \epsilon \sim \mathcal{N}(0, 3)$.

#### S2.1.3 Impact of the evidential regularizer

In the following experiment, we demonstrate the importance of augmenting the training objective with our evidential regularizer $\mathcal{L}^R$ as introduced in Sec. 3.3. Fig. S2 provides quantitative results on epistemic uncertainty estimation after training on the same regression problem presented in S2.1 with different realizations of the regularization coefficients, $\lambda$. We show the performance of our ability to calibrate uncertainty on OOD data is heavily related to our regularizer. As we decrease our regularizer weight, uncertainty on OOD examples decays to zero. Stronger regularization inflates the uncertainty ($\lambda = 0.01$ is a good choice for this problem) while aleatoric uncertainty is maintained constant. Please refer to Fig. 3 for the regularization effect on both aleatoric and epistemic uncertainty.

#### S2.1.4 Disentanglement of aleatoric and epistemic uncertainty

In the following experiment, we provide results to suggest that the evidential regularizer is capable of disentangling aleatoric and epistemic uncertainties by capturing incorrect evidence. Specifically,

15

Figure S2: **Impact of regularization strength on epistemic uncertainty estimates.** Epistemic uncertainty estimates on the dataset $y = x^3 + \epsilon, \epsilon \sim \mathcal{N}(0, 3)$ for evidential regression models regularized with the evidential regularizer $\mathcal{L}^R$ (A) or with the KL divergence (B) between the inferred NIG and another with zero evidence, for varying regularization coefficients $\lambda$.

we construct a synthetic toy dataset with high data noise (aleatoric uncertainty) in the center of the in-distribution region. Rather than using the $L1$ error in the regularization term, as in previous experiments, we use regularize the standard score and estimate epistemic and aleatoric uncertainty (Fig. S3). This analysis suggests that the method is capable of disentangling epistemic and aleatoric uncertainties in a region that is in-distribution but has high data noise.

Figure S3: **Disentangled uncertainties.** Epistemic and aleatoric uncertainty estimates on a synthetic dataset based on $y = x^3$, where data noise increases towards the center of the in-distribution region. The evidential regularizer $\mathcal{L}^R$ is calculated based on the standard score.

### S2.2 Benchmark regression problems

#### S2.2.1 Datasets and experimental setup

This subsection describes the setup to create Table 1. We follow an identical experimental setup and training process as presented in [20]. All dataset features are normalized to have zero mean and unit standard deviation. Features with no variance are only normalized to have zero mean. The same normalization process is also performed on the target variables; however, this is undone at

16

inference time such that predictions are in the original scale of the targets. Datasets are split randomly into training and testing sets a total of 20 times. Each time we retrain the model and compute the desired metrics (RMSE, NLL, and speed). The results presented in the table represent the average and standard error across all 20 runs for every method and dataset. Following the lead of [28], we also directly compare against the other training methods by directly using their reported results since they followed an identical training procedure.

## S3 Depth estimation evaluations

### S3.1 Experimental details

We evaluate depth estimation on the NYU-Depth-v2 dataset [35]. For every image scan in the dataset we fill in the missing holes in the depth using the Levin Colorization method. The resulting depth map is converted to be proportional to disparity by taking its inverse. This is common in depth learning literature as it ensures that far away objects result in numerically stable neural network outputs (very large depths have close to zero disparity). Objects closer than 1/255 meters to the camera would therefore be clipped due to the `uint8` restriction on image precision. The resulting images are saved and used for supervising the learning algorithm. Training, validation, and test sets were randomly split (80-10-10) with no overlap in scans.

All trained depth models have a U-Net [41] backbone, with five convolutional and pooling blocks down (and then back up). The input and target images had shape $(160, 128)$ with inputs having 3 feature maps (RGB), while targets only had a single feature map (disparity). The dropout variants were trained with spatial dropout [45] over the convolutional blocks ($p = 0.1$). Evidential models additionally had four output target maps, one map corresponding to each evidential parameter $\gamma, \upsilon, \alpha, \beta$, with activations as described in 3.3.

All models were trained with the following hyperparmeters: batch size of 32, Adam optimization with learning rate 5e-5, over 60000 iterations. The best model according to validation set RMSE is saved and used for testing. Evidential models additionally had $\lambda = 0.1$. Each model was trained 3 times from random initialization to produce all presented results.

### S3.2 Depth estimation performance metrics

Table S1 summarizes the size and speed of all models. Evidential models contain significantly fewer trainable parameters than ensembles (where the number of parameters scales linearly with the size of the ensemble). Since evidential regression models do not require sampling in order to estimate their uncertainty, their forward-pass inference times are also significantly more efficient. Finally, we demonstrate comparable predictive accuracy (through RMSE and NLL) to the other models.

|  | N | # Parameters | | Inference Speed | | RMSE | NLL |
|---|---|---|---|---|---|---|---|
|  |  | Absolute | Relative | Seconds | Relative |  |  |
| **Evidential (Ours)** | - | 7,846,776 | 1.00 | 0.003 | 1.00 | $0.024 \pm 0.032$ | $-1.128 \pm 0.290$ |
| **Spatial Dropout** | 2 | 7,846,657 | 1.00 | 0.028 | 10.20 | $0.033 \pm 0.037$ | $-0.564 \pm 0.231$ |
| **Spatial Dropout** | 5 | 7,846,657 | 1.00 | 0.031 | 11.48 | $0.031 \pm 0.033$ | $-1.227 \pm 0.374$ |
| **Spatial Dropout** | 10 | 7,846,657 | 1.00 | 0.037 | 13.69 | $0.035 \pm 0.042$ | $-1.139 \pm 0.379$ |
| **Spatial Dropout** | 25 | 7,846,657 | 1.00 | 0.065 | 23.99 | $0.032 \pm 0.035$ | $-1.137 \pm 0.327$ |
| **Spatial Dropout** | 50 | 7,846,657 | 1.00 | 0.107 | 39.36 | $0.032 \pm 0.036$ | $-1.110 \pm 0.381$ |
| **Ensembles** | 2 | 15,693,314 | 2.00 | 0.005 | 1.94 | $0.026 \pm 0.032$ | $-1.080 \pm 3.334$ |
| **Ensembles** | 5 | 39,233,285 | 5.00 | 0.010 | 3.72 | $0.023 \pm 0.027$ | $-1.077 \pm 0.298$ |
| **Ensembles** | 10 | 78,466,570 | 10.00 | 0.019 | 6.82 | $0.025 \pm 0.038$ | $-0.980 \pm 0.298$ |
| **Ensembles** | 25 | 196,166,425 | 25.00 | 0.045 | 16.45 | $0.022 \pm 0.029$ | $-1.000 \pm 0.259$ |
| **Ensembles** | 50 | 392,332,850 | 50.00 | 0.112 | 41.26 | $0.022 \pm 0.031$ | $-0.996 \pm 0.275$ |

Table S1: **Depth estimation performance metrics.** Comparison of different uncertainty estimation algorithms and predictive performance on an unseen test set. Dropout and ensembles were sampled N times on parallel threads. The evidential method outperforms all other algorithms in terms of space (#Parameters) and inference speed while maintaining competitive RMSE and NLL.

### S3.3 Epistemic uncertainty estimation on depth

Fig. S4 shows individual trial runs for each method on RMSE cutoff plots as summarized in Fig. 4B.

17

Fig. S5 shows individual trial runs for each method on their respective calibration plots as summarized in Fig. 4C.

Fig. S6 shows individual trial runs for each method on their respective entropy (uncertainty) CDF as a function of the amount of adversarial noise. We present the evidential portion of this figure in Fig. 6C, but also provide baseline results here.

Figure S4: **Relationship between prediction confidence level and observed error for different uncertainty estimation methods.** A strong inverse trend is desired to demonstrate that the uncertainty estimates effectively capture accuracy. Plots show results from depth estimation task.

Figure S5: **Uncertainty calibration plots for depth estimation.** Calibration of epistemic uncertainty estimates for dropout, ensembling, and evidential methods, assessed as the relationship between expected and observed predictive confidence levels. Perfect calibration corresponds to the line $y = x$ (black).

Figure S6: **Effect of adversarial noise on uncertainty estimates**. Cumulative distribution functions (CDF) of entropy (uncertainty) estimated by dropout, ensembling, and evidential regression methods, under the presence of increasing adversarial noise $\epsilon$.

### S3.4 Aleatoric uncertainty estimation on depth

Fig. S7 compares the evidential aleatoric uncertainty to those obtained by Gaussian likelihood optimization in several domains with high data uncertainty (mirror reflections and poor illumination). The results between both methods are in strong agreement, identifying mirror reflections and dark regions without visible geometry as sources of high uncertainty. These results are expected since evidential models fit the data to a higher-order Gaussian distribution and therefore it is expected that they can accurately learn aleatoric uncertainty (as is also shown in [42, 18]). While the main text

18

Figure S7: **Aleatoric uncertainty in depth.** Visualizing predicted aleatoric uncertainty in challenging reflection and illumination scenes. Comparison between evidential and [25] show strong semantic agreement.

focuses on the more challenging problem of epistemic uncertainty estimation (especially on OOD data), we provide these sample aleatoric uncertainty examples for here for depth as supplemental material.



We thank the reviewers for their very constructive and detailed feedback on our manuscript. In this work, we propose a novel and scalable method for inferring a continuous target as well as representations for epistemic and aleatoric uncertainty, without sampling during inference. Our method does not require any out-of-distribution (OOD) data during training (unlike Dirichlet Prior Networks [24]) and performs on-par with or better than state-of-the-art (SoA) approaches. We demonstrate learning well-calibrated measures of uncertainty on various benchmarks, scaling to high-dimensional vision tasks, as well as robustness to new OOD and adversarial test samples. As a sampling-free and performant method, our work will enable key advances in resource constrained areas, such as robotics, where sampling is infeasible.

Fig. 1. Disentangled uncertainty

**R1:** *3.1. Pseudo-counts:* The overall evidence is presented as the sum of pseudo-counts [32]. We could equivalently average, by applying a constant re-scaling directly captured by $\lambda$, without changing any of our results. *3.2a. Regularizer and results:* We agree that our method provides no guarantees that it will definitively yield high epistemic uncertainty far from in-domain regions; however, we believe that the extensive empirical results achieved with our method, and results from similar related approaches which also train on *only in-distribution data* (i.e., [32], [Joo, T. et al. '20]), support the claim that uncertainty increases on OOD data. Our approach will undoubtedly improve by leveraging the supervision of OOD data during training, closer to [24, 3]; however, as we (and R2) point out, the need for OOD training data is often a critically limiting assumption. *3.2b. "Confused evidence":* As R1 correctly states, the regularizer captures scenarios where the evidence is leading to the incorrect target (i.e., incorrect or "confused" evidence, not lack of evidence). We fully agree with this excellent point and have updated the manuscript to reflect this. However, we do not believe that the approach "conflates aleatoric and epistemic uncertainty" and provide results from the suggested experiment to support our claim (Fig. 1), using the *standard score* instead of L1 error. Further details and analysis are added to the manuscript. *3.3. Other metrics:* Leveraging evidential distributions to compute M.I. or even differential entropy is a great idea, as these are rich statistics that our method captures. We focus on first order moments for more direct comparability to existing SoA baselines and leave further analysis of richer statistics to future work. *4.1. Performance:* RMSE for our method (and baselines) is in fact provided in Tab. S1, Figs. 4B, 6A. We observed little to no performance loss based on RMSE and will certainly include the other metrics as suggested. *4.2. AUC:* The histograms (and CDFs) provided in Figs. 5, 6, and S5 (as in [21], [Nalisnick, E. et al. '18], and others) are richer performance statistics and directly reduce to the requested AUC-ROC scores. To address these concerns, we have added all AUC-ROC values to our performance charts. *4.3 Adversarial:* We updated the implementation details of the attack method (FGSM). While we can evaluate additional attacks, our paper is *not* proposing a new defense (neither are any of our baselines), and thus this would be out of scope. The goal of Sec. 4.3.1 is solely to evaluate on additional OOD samples based on a basic adversarial method.

**R2:** *1. Fig. 3 aleatoric:* Within the training region there are very few differences, which can be attributed to intrisinic randomness and initialization during training. OOD there is much more variability, aligning with MVE [18, 28]. Since there is no training data in this region, we do not expect consistent results for aleatoric uncertainty, unlike epistemic uncertainty as is pointed out. *2. Relation to Kendall. et al:* This is an excellent and very important point; we apologize for the confusion. To clarify, estimating aleatoric uncertainty using NNs without sampling has a well-accepted solution dating back to 1994 with MVE (see [28]). This is the same approach used in Kendall et al. [18] and is what we compare against in our work (Fig. 3, and elsewhere when evaluating aleatoric uncertainty). Further, [18] proposes jointly learning MVE with a sampling-based epistemic uncertainty estimator (in their paper, dropout [5]). Thus, in [18] aleatoric can be estimated sampling-free, but epistemic requires sampling. We believe all our provided benchmarks do indeed accurately compare against [18]. The majority of our results focus on epistemic comparisons since our method uses a Gaussian lower-order distribution which achieves aleatoric estimation similar to [18] using MLE. In contrast, we jointly learn a *sampling-free* epistemic estimate which is not the case in [18], representing a key contribution of our work. Sampling approaches, including [18], are the current SoA and we agree with R2 that the benchmarking we provide on these methods is absolutely critical. *3. Intuition of parameters:* Thank you, additional exposition has been added. *4. Baselines:* Please refer to #2 above, which clarifies the incorrect point about missing baselines.

**R3:** *1. Gaussian assumption:* Thank you, explanations will be added. *2. Additional comparisons and prior work:* Discussion on these works will be added. Note, [20] proposes a way to calibrate a given uncertainty method as opposed to a new uncertainty estimator, and can be used to strengthen any uncertainty estimator - it is not a competing method.

**R4:** *1. Other distributions:* Excellent point to be included in the manuscript. *2. Performance on OOD:* Results for a variant of the proposed experiment can be found in Fig. 5. Further metrics have also been added via R1 #4.2.

**Summary:** Thank you for running our software. R3: *"This is so far the only code I was able to run among the ones I have to review. Authors really went to a great length to provide runable code, and this is commendable."* We believe this work supports new research through its broad applicability and accessible, easy to use code. We hope the rebuttal and new experiments address all concerns (esp. R1), and appreciate all comments which have improved the manuscript.