[Reviews · NeurIPS 2020]

Review 1

Summary and Contributions: This work proposes an evidence based approach to model both epistemic and aleatoric uncertainty in regression tasks using a single model. They propose to parameterise a Normal inverse-Gamma distribution, which is a distribution over normal distributions, using neural network. This model is then trained with a standard maximum likelihood loss with an additional "evidence regularisation" term. The proposed approach is competitive with ensemble methods on a range of tabular datasets, as well as on the NYU Depth V2 monocular depth estimation task.

Strengths: This paper is relevant to the NeurIPS community. The author's approach is sensible and essentially parallels Dirichlet Prior Networks, but for regression tasks. Results on the tabular UCI datasets are promising and show that the proposed approach is at least comparable to established ensemble-based techniques. Similarly, the results on the monocular depth estimation task also show promise. However, I have several reservations.

Weaknesses: While the overall approach is very sensible, I have several theoretical reservations. 1. If both nu and 2*alpha are pseudo-counts of prior observations, then why are we defined overall evidence as 2*alpha + nu? Why not the average of the two? Conceptually, it makes little sense to predict a different number of pseudo-counts for the priors over the mean and the variance. Ultimately, this is absorbed into gamma, the regularisation weight, but nevertheless. 2. I am not convinced that the proposed evidence regularizer really does capture epistemic uncertainty or lack of evidence far away from the training data, despite the results. There are two reasons for this. Firstly, the model is exposed *exclusively* to in-domain data during training, and since the authors are doing a form of MAP estimation (MLE + regulariser) there is absolutely no guarantee that far from the in-domain region the model will actually yield high estimates of epistemic uncertainty - there is no data and no structural properties (RBF functions, Gaussian Process output layer or ensembles) which ensure this. The model *may* generalise the effects of the regulariser to a certain region beyond the in-domain training data, but there is no way to guarantee this behaviour. Note, that similar concern's can be addressed to the evidential approach to Dirichlet Prior Networks proposed in Sensoy et al. I suggest that the authors attempt a counter-factual experiment - rather than showing that near the edge of the OOD region the variance increases, show that it *does not decrease* with increasing distance from the in-domain region. Secondly, I am also not convinced that the proposed regulariser actually captures a *lack of evidence* rather than *confused evidence*. Note how similar the regularizer is (absolute error * evidence) to the MSE term in the likelihood for a standard gaussian (MSE * precision / 2). This suggests that before the models are converged, variance (or nu + 2*alpha, in this case) will be large in regions with a large error. As the models are trained further, the variance will naturally capture regions of *conflicting evidence*, where the expected squared error is larger due to conflicting targets, rather than *lack of evidence*. Thus, I believe the proposed approach will conflate aleatoric and epistemic uncertainty. This is a large concern which the authors do not address. A counter-factual for this would be to construct a dataset with a very high level of noise (aleatoric uncertainty) at the centre of the in-domain region, and then to show that epistemic uncertainty is nevertheless zero in this region. 3. Why are only variance of mean and expected variance considered as measure of uncertainty. Given that the Normal-inverse-Wishart is also a member of the exponential family, why not consider the standard information-theoretic decomposition of uncertainty via mutual information? This may provide more informative measures of uncertainty than variances, as this captures higher moments. 4. Methodological concerns are addressed below.

Correctness: Methodologically, I have concerns regarding the monocular depth-estimation experiments. 1. Firstly, the base-performance of the systems are not given in terms of RMSE, detla_1, delta_2 and delta_3 , log-error and relative error, as is standard in depth-estimation literature. This is important, as it is not standard to train *probabilistic* depth estimation models (most current SOTA cocktail of bespoke losses for each task). If the method degrades performance relative to standard approaches, it is unlikely to be used. 2. Secondly, OOD detection is typically assess via AUC-ROC (not given) relative to a range of OOD datasets. In this work only a single OOD dataset was considered. Comparison of OOD detection relative to baselines is also not provided. 3. It is not clear what kind of adversarial attack is used. Furthermore, robustness to adversarial examples should be evaluated relative to *adaptive* or *uncertainty aware* attacks. I understand that the authors in this case are using "standard" (likely FGSM) adversarial attacks as another kind of OOD dataset, but this has been shown to be a very simple attack to detect.

Clarity: The paper is clearly written, well illustrated and easy to understand.

Relation to Prior Work: I believe that this work correct cites prior work.

Reproducibility: Yes

Additional Feedback: --- POST REBUTTAL COMMENTS --- I think the authors have addressed a range of my critiques - mainly that they are able to distinguish aleatoric and epistemic uncertainty as well as the performance figures and AUC-ROCS. At this point I'm willing to up the score to a 5, acknowledging the author's consistent and admirable effort and that MOST of the proposed method IS very sensible. However, I still have strong concerns regarding the regulariser (and the authors acknowledge this ) - it is surprising to me that a regulariser which forces the model to predict *low evidence* in areas of *conflicting evidence* yields *low evidence* in areas of *low evidence*. There seems to be no theoretically justification for this to work. Furthermore, the proposed method does not yield any guarantees, either via data or via model structure, that it will yield low evidence in OOD regions (and the authors acknowledge this as well). Thus, we essentially have a method which seems to empirically work due to a serendipitous coincidence, but not theoretical insights into why it works. Given that the aim of uncertainty estimation is to improve *safety*, I am hesitant to recommend acceptance of a method which works for poorly understood (to my mind) reasons and offers no guarantees. I strongly encourage the authors to explore in greater depth the mechanism behind their method. For example, does the model make greater errors in areas of low evidence? Or this a properly of under-fitting? Does the model retain its uncertainty estimation properties if it is trained for far longer? Can it still separate aleatoric and epistemic uncertainty if it is trained for longer? These are some questions which can guide the authors.


Review 2

Summary and Contributions: - Paper proposes a novel method for training non-Bayesian NNs to estimate a continuous target as well as its associated evidence in order to learn both aleatoric and epistemic uncertainty - No need for multiple passes and/or OOD datasets during training - The paper formulates an evidential regularizer for continuous regression problems - evidential deep learning is been designed specifically for classification, this work looks into the regression setting. - The goal is to avoid multiple passes (needed by BNNs) and only learn from in-distribution data (because cant easily specify what is OOD). - In order to address the issue of assign higher uncertainty for incorrect predictions, the paper introduces a novel evidence regularizer. Previously suggested solutions mainly focus on the classification setting and the simple regression equivalent of the solution is undefined.

Strengths: - Very well written paper and the theory is nicely explained and correct - I specifically like that almost every decision or design choice has been justified or discussed - Results in Tab. 1 seem to show mostly similar performance to sampling-based alternatives though as the proposed method is sampling-free the method is much quicker. This is a practical advantage of the method compared to the compared alternatives. - Qualitative results look convincing compared to the two baselines. Specifically, I liked the aleatoric uncertainty maps in the appendix. Though, I still feel the comparison against the aleatoric attenuation loss from Kendal & Gall 2017 is still missing and as I cant compare against this it is hard to tell if this method improves the quality (see weaknesses).

Weaknesses: - Fig. 3 (Toy example): the advantage of the proposed method for estimating epistemic uncertainty seems to be clear for OOD data. Though I am struggling to see why and how the proposed method is better at aleatoric uncertainty estimation. There also seems to be not much discussion about this other than stating that all methods "accurately capture uncertainty within the training distribution". I think this needs some more discussion as currently I do not see the benefit of the estimated aleatoric uncertainty. Also, what is causing the slight differences in the aleatoric uncertainty estimates for the three methods? For the baseline, does that really capture the aleatoric uncertainty? It seems that for the OOD regions, the uncertainty can and will be reduced with more data, though shouldnt this type of uncertainty be irreducible by more data? - There seems to be no comparison against the original work introducing disentanglement of aleatoric and epistemic uncertainty [18 from paper bib.]. Specifically, the aleatoric estimate. The paper is cited and the distenglements of the uncertainties introduced by this paper are also used, but there is no comparision against it. They introduced the aleatoric loss (in addition to estimating the uncertainty, they showed that this loss formulation also helps improve the accuracy of the models) and this paper seems to be incomplete without a comparison against this, therefore putting the results into context seems to be hard. Using the aleatoric loss would also result in a sampling-free method, so the comparison against this is of utmost importance, as the "speed" benefits argument which the paper promotes only holds for the two compared slow methods (MC-Dropout and Ensembles). When comparing against sample-free approaches, the benefit of speed would not be there anymore and then have merely similar performance would not be enough to be a competitive technique. There is no comparison against any sample-free approach. In summary, the paper is missing an important comparison which the authors know about (i.e. they cited the work) though still failed to compare against. As there are no other sampling-free methods compared against, I find it hard to believe the benefits of this method. So far the only benefit is that it is faster than two (relatively weak) notoriously slow methods. - As someone unfamiliar with Evidential Regression, it seems hard to follow and understand the intuition behind the parameters of the High-order distribution. Makes it hard to easily interpret the different equations when the parameters and the influence they have is unknown. It required some looking into before it become slightly more clear. - Interesting work, though it seems this paper mainly focuses on addressing the issues which arise when applying previous works, designed for classification, to regression. Therefore, I feel the novelty of the work can be put into question, though I still think the authors did a good job addressing this. So I would not weight this weakness too high. - Even though the investigations into the behaviour of the methods on OOD data is interesting, the results do not seem to be very convincing. Owing to these similar performances across the 3 methods, it seems the authors also haven't discussed this results much. For example, in Fig. 5b: "Fig. 5B summarizes these entropy distributions as interquartile boxplots to again show clear separation in the uncertainty distribution on OOD data." This statement holds for all 3 methods. Already there are missing comparison methods and then again for this task, the performance is not much different either. - Similar comments on the "Robustness to adversarial samples". It is hard to see the benefit of this method compared to the baselines. Overall the results are not always convincing except for the fact that it's faster than the other 2 notoriously slower approaches.

Correctness: Yes

Clarity: Yes

Relation to Prior Work: Yes

Reproducibility: Yes

Additional Feedback: ---------------- Review Update ---------------------- I thank the authors for addressing some of my concerns. The major concern about the comparison and baselines has been clarified for me. I will up my score from the original 6 to 7. I still have some concerns about the novelty of the work (i.e. extension from classification to regression), though at the same time admit that the authors did a good job for this "type" of paper. The experiment section could still be further improved and the gains of the method can be put into question. Though I still think in its current form the paper is good enough.


Review 3

Summary and Contributions: The paper proposes to learn a end-to-end regression network that also estimate the output uncertainty in a Bayesian fashion. This is done by assuming normality of the error and by including in the loss functions the parameters of classical Bayesian prior for normal distributions. The framework is tested against dropout and ensemble techniques in a number of scenarii (depth estimation, OOD samples, adversarial noise), each time with one data set. The provided results show an improvment with respect to dropout, but are roughly on a par with ensemble techniques.

Strengths: * An easy to implement method inspired from Bayesian approaches to assess uncertainty in regression problems. * Experiments on a number of scenarii demonstrating the applicability and good (even if not mind blowing) performances of the proposed method.

Weaknesses: * A more in-depth discussion about the validity of the assumption (authors assume a Gaussian distribution, but how realistic is it?). * Some more comparisons with other frameworks that intend to assess uncertainties in prediction and to provide calibrated assessment of uncertainties.

Correctness: * The developments in the paper are clear enough as well as the experiments. Additionally, this is so far the only code I was able to run among the ones that have been joined to papers I have to review. Authors really went to a great length to provide runable code, and this is commendable.

Clarity: * Yes it is

Relation to Prior Work: I do miss some discussion as well as comparative experiments with frameworks having the same intent: * a first is one is the approach developed in "Kuleshov, Volodymyr, Nathan Fenner, and Stefano Ermon. "Accurate Uncertainties for Deep Learning Using Calibrated Regression." ICML. 2018." * A second one are conformal methods, that can be plugged in into any regressor (including NN) and gives calibrated predictions (with the width of such predictions quantifying the uncertainty amount) without much efforts. Why not considering them rather than developing new methods? A possible reference is "Harris Papadopoulos and Haris Haralambous. Reliable prediction intervals with regression neural networks. Neural Networks, 24(8):842–851, 2011" I also do not understand why the approach in reference [20] (cited to compute calibration curves) is not discussed?

Reproducibility: Yes

Additional Feedback:


Review 4

Summary and Contributions: The paper presents an evidence-based prior on the likelihood function for modeling uncertainty in regression of continuous variables. Rather than the Bayesian NN's use priors on network weights, this approach side-steps the need to perform sampling. At the output, the proposed evidential regression produces estimates of both aleatoric and epistemic uncertainty that are well-calibrated compared with competing methods. This prior serves as regularization that calibrates uncertainty on out-of-distribution (OOD) cases. Post-rebuttal: Despite the unanswered questions, I felt the paper does a good job at motivating and describing an approach, and performs a solid set of experiments on real-world examples. While not all of the theoretical questions are answered, I feel it's enough to comfortably warrant acceptance, and that conference attendees would be interested, perhaps even facilitating others to take the next steps to close some of the identified gaps. I maintain my rating as "8".

Strengths: Positives: The paper is well-written and follows a logical development of the presentation of the background, the algorithm formulation, and the experimental results. Results are shown on a variety of problems including toy problems, OOD situations, and adversarial OOD cases. The paper is well-motivated and as far as I know, this presentation of evidential priors for regression tasks is novel.

Weaknesses: The paper's presented evidential prior is based on a Normal Inverse Gamma -- it would be beneficial to discuss extensions to other distributions. e.g., extensions to distributions with multiple modes. Overall, experiments capture the flexibility of the approach. However, perhaps the performance on OOD inputs could be quantified. For example, when an input image from another dataset is presented, can the epistemic uncertainty classify this as such? E.g., Nalisnick et al: Do Deep Generative Models Know What They Don't Know? ICLR 2019 and others quantify OOD classification when training on one dataset and testing on an OOD set. Perhaps a variant of this could be performed, e.g., by training a regression model on MNIST, then presenting an SVHN image.

Correctness: The presented claims appear to be correct.

Clarity: Yes, the paper is carefully written, well organized, and well motivated. The algorithm development is clear, and mathematical notation is clear and consistent.

Relation to Prior Work: Yes, the paper draws contrasts and comparisons with other works throughout its exposition. In particular, the paper does a nice job at placing its contribution with respect to Bayesian NNs such as [18].

Reproducibility: Yes

Additional Feedback:

[Author Response · NeurIPS 2020]

We thank the reviewers for their very constructive and detailed feedback on our manuscript. In this work, we propose a novel and scalable method for inferring a continuous target as well as representations for epistemic and aleatoric uncertainty, without sampling during inference. Our method does not require any out-of-distribution (OOD) data during training (unlike Dirichlet Prior Networks [24]) and performs on-par with or better than state-of-the-art (SoA) approaches. We demonstrate learning well-calibrated measures of uncertainty on various benchmarks, scaling to high-dimensional vision tasks, as well as robustness to new OOD and adversarial test samples. As a sampling-free and performant method, our work will enable key advances in resource constrained areas, such as robotics, where sampling is infeasible.

**R1:** *3.1. Pseudo-counts:* The overall evidence is presented as the sum of pseudo-counts [32]. We could equivalently average, by applying a constant re-scaling directly captured by $\lambda$, without changing any of our results. *3.2a. Regularizer and results:* We agree that our method provides no guarantees that it will definitely yield high epistemic uncertainty far from in-domain regions; however, we believe that the extensive empirical results achieved with our method, and results from similar related approaches which also train on *only in-distribution data* (i.e., [32], [Joo, T. et al. '20]), support the claim that uncertainty increases on OOD data. Our approach will undoubtedly improve by leveraging the supervision of OOD data during training, closer to [24, 3]; however, as we (and R2) point out, the need for OOD training data is often a critically limiting assumption. *3.2b. "Confused evidence":* As R1 correctly states, the regularizer captures scenarios where the evidence is leading to the incorrect target (i.e., incorrect or "confused" evidence, not lack of evidence). We fully agree with this excellent point and have updated the manuscript to reflect this. However, we do not believe that the approach "conflates aleatoric and epistemic uncertainty" and provide results from the suggested experiment to support our claim (Fig. 1), using the *standard score* instead of L1 error. Further details and analysis are added to the manuscript. *3.3. Other metrics:* Leveraging evidential distributions to compute M.I. or even differential entropy is a great idea, as these are rich statistics that our method captures. We focus on first order moments for more direct comparability to existing SoA baselines and leave further analysis of richer statistics to future work. *4.1. Performance:* RMSE for our method (and baselines) is in fact provided in Tab. S1, Figs. 4B, 6A. We observed little to no performance loss based on RMSE and will certainly include the other metrics as suggested. *4.2. AUC:* The histograms (and CDFs) provided in Figs. 5, 6, and S5 (as in [21], [Nalisnick, E. et al. '18], and others) are richer performance statistics and directly reduce to the requested AUC-ROC scores. To address these concerns, we have added all AUC-ROC values to our performance charts. *4.3 Adversarial:* We updated the implementation details of the attack method (FGSM). While we can evaluate additional attacks, our paper is *not* proposing a new defense (neither are any of our baselines), and thus this would be out of scope. The goal of Sec. 4.3.1 is solely to evaluate on additional OOD samples based on a basic adversarial method.

Fig. 1. Disentangled uncertainty

**R2:** *1. Fig. 3 aleatoric:* Within the training region there are very few differences, which can be attributed to intrinsic randomness and initialization during training. OOD there is much more variability, aligning with MVE [18, 28]. Since there is no training data in this region, we do not expect consistent results for aleatoric uncertainty, unlike epistemic uncertainty as is pointed out. *2. Relation to Kendall. et al:* This is an excellent and very important point; we apologize for the confusion. To clarify, estimating aleatoric uncertainty using NNs without sampling has a well-accepted solution dating back to 1994 with MVE (see [28]). This is the same approach used in Kendall et al. [18] and is what we compare against in our work (Fig. 3, and elsewhere when evaluating aleatoric uncertainty). Further, [18] proposes jointly learning MVE with a sampling-based epistemic uncertainty estimator (in their paper, dropout [5]). Thus, in [18] aleatoric can be estimated sampling-free, but epistemic requires sampling. We believe all our provided benchmarks do indeed accurately compare against [18]. The majority of our results focus on epistemic comparisons since our method uses a Gaussian lower-order distribution which achieves aleatoric estimation similar to [18] using MLE. In contrast, we jointly learn a *sampling-free* epistemic estimate which is not the case in [18], representing a key contribution of our work. Sampling approaches, including [18], are the current SoA and we agree with R2 that the benchmarking we provide on these methods is absolutely critical. *3. Intuition of parameters:* Thank you, additional exposition has been added. *4. Baselines:* Please refer to #2 above, which clarifies the incorrect point about missing baselines.

**R3:** *1. Gaussian assumption:* Thank you, explanations will be added. *2. Additional comparisons and prior work:* Discussion on these works will be added. Note, [20] proposes a way to calibrate a given uncertainty method as opposed to a new uncertainty estimator, and can be used to strengthen any uncertainty estimator - it is not a competing method.

**R4:** *1. Other distributions:* Excellent point to be included in the manuscript. *2. Performance on OOD:* Results for a variant of the proposed experiment can be found in Fig. 5. Further metrics have also been added via R1 #4.2.

**Summary:** Thank you for running our software. R3: *"This is so far the only code I was able to run among the ones I have to review. Authors really went to a great length to provide runable code, and this is commendable."* We believe this work supports new research through its broad applicability and accessible, easy to use code. We hope the rebuttal and new experiments address all concerns (esp. R1), and appreciate all comments which have improved the manuscript.

[Meta-Review · NeurIPS 2020]

The authors present a way to improve uncertainty quantification in regression networks by placing "evidential priors" over the conditional distributions and optimizing the marginal likelihood. Pros: - simplicity and elegance of the approach - experimental results showing that this can match SOTA (deep ensemble) performance with lower computational complexity at test time - Code accompanying the paper that allows the community to easily build on this work Cons: - Theoretical justification could be improved. In particular, R1 raised some concerns about why this works. Some of the ablations in rebuttal partly address this concern. I'd also encourage the authors to include additional ablations to investigate the relative contributions of (i) using prior over Gaussian parameters and (ii) details of how these parameters are optimized (e.g. maximum likelihood vs the proposed regularization). During the discussion, the consensus leaned towards accept. I have read the paper as well and I think this is an useful contribution. I recommend accept.